# Symbiotic fouling of *Vetulicola*, an early Cambrian nektonic animal

Yujing Li [1,2,3 ✉], Mark Williams [2,4], Thomas H. P. Harvey[2,4], Fan Wei[1,2], Yang Zhao [1,2], Jin Guo[1,2,5], Sarah Gabbott[2,4], Tom Fletcher[2,4], Xianguang Hou[1,2] & Peiyun Cong [1,2,3 ✉]

Here, we report the earliest fossil record to our knowledge of surface fouling by aggregates of small vermiform, encrusting and annulated tubular organisms associated with a mobile, nektonic host, the enigmatic Cambrian animal *Vetulicola*. Our material is from the exceptionally preserved early Cambrian (Epoch 2, Age 3), Chengjiang biota of Yunnan Province, southwest China, a *circa* 518 million-year old marine deposit. Our data show that symbiotic fouling relationships between species formed a component of the diversification of animal-rich ecosystems near the beginning of the Phanerozoic Eon, suggesting an early escalation of intimate ecologies as part of the Cambrian animal radiation.

[1] Yunnan Key Laboratory for Palaeobiology, Institute of Palaeontology, Yunnan University, Kunming, Yunnan 650500, China. [2] MEC International Joint Laboratory for Palaeobiology and Palaeoenvironment, Institute of Palaeontology, Yunnan University, Kunming, Yunnan 650500, China. [3] State Key Laboratory of Palaeobiology and Stratigraphy, Nanjing Institute of Geology and Palaeontology, CAS, Nanjing, Jiangsu 210008, China. [4] Centre for Palaeobiology Research, School of Geography, Geology and the Environment, University of Leicester, Leicester LE1 7RH, UK. [5] Management Committee of the Chengjiang Fossil Site World Heritage, Chengjiang 652599, China. ✉email: yujingli@ynu.edu.cn; cong@ynu.edu.cn

The early Cambrian was a time of major evolutionary change with the development of complex animal-diverse marine ecosystems. Biotic interactions are considered to have played a major role in driving taxonomic diversification and ecological escalation through this interval, although direct fossil evidence is rare. The coevolution of predators and prey has received particular attention, with fossil support from gut contents and faecal pellets, and indirectly, from adaptations for prey capture or defence from predators[1,2]. In contrast, symbiotic interactions, such as mutualism and parasitism, are difficult to diagnose from fossils, and are extremely rare in the Cambrian[3,4].

Here, we identify a new symbiotic component to Cambrian marine ecologies, in which worm-like animals are preserved attached to a mobile host, the enigmatic animal *Vetulicola*. Furthermore, the attaching organisms are almost all preserved *inside* the mouldic vetulicolian fossils, indicating that they were attached to inner surfaces. Hosts show no sign of decay and disarticulation, host and symbiont are similarly preserved and symbionts are not found anywhere else, indicating that the symbiosis was very specific and occurred in life. The robust tubular forms of the inhabitants suggest a sedentary surface-encrusting habit rather than invasion of the soft tissues. The number of inhabitants in some specimens (>45) may have induced a negative effect on the host, the attaching endosymbionts partly obstructing water flow, i.e., biofouling of internal body surfaces.

## Results

### Systematic palaeontology. Clade Bilateria, Clade Protostomia

*Vermilituus gregarius* gen. et sp. nov.

*Etymology*: Genus name from *vermis* (Latin) meaning worm and *lituus* (Latin) meaning a curved trumpet, alluding to the shape of the fossils. Species name from *gregarius* (Latin), meaning flock or herd.

*Holotype*: YKLP 13079a, b (counterparts), U-shaped tube (Fig. 1a, b), 6.5-mm long, and reaching a maximum width of 0.6 mm: the holotype is associated with *Vetulicola rectangulata* YKLP 13075a, b. Paratypes (with preserved shell annulation), YKLP 13084 and 13085 (Fig. 1c, f) associated with *V. rectangulata* YKLP 13074, and YKLP 13082 and 13083 (Fig. 1e) associated with *V. rectangulata* YKLP 13073.

*Referred material*: About 192 specimens from Ercaicun, 75 from Mafang and 10 from Jianshan associated with *Vetulicola rectangulata*, all in the collections of the Yunnan Key Laboratory for Palaeobiology (YKLP). In total, 17 specimens from Xiaolantian and 55 specimens from Heimadi associated with *Vetulicola cuneata*, all in the collections of the Chengjiang Fossil Museum (CJHMD, Supplementary data file).

*Locality*: Ercaicun (type locality), Mafang and Jianshan localities in the Haikou area of Kunming, and Xiaolantian and Heimadi in Chengjiang County, Yunnan Province, China (for localities see ref. [5]).

*Horizon*: Yu'anshan Member, Chiungchussu Formation, *Eoredlichia-Wutingaspis* trilobite Biozone, Nangaoan Stage of Chinese regional usage, Cambrian Series 2, Stage 3. All specimens are from rapidly sedimented 'event beds'[5].

*Diagnosis for genus (monotypic) and species*. Small (0.8–7.2-mm long) elongated, conical tubes having three general forms, as a U-shape, J-shape or complex sinusoid, the latter being the dominant type: occasionally the tube also begins with a 360° planispiral coil before straightening. Coiling can be both dextral and sinistral and is in a single plane. The proximal end of the tube blunts (no bulb-like origin). Tubes increase in diameter very slowly, the proximal diameter being about 0.2 mm and the distal diameter reaching 1 mm. No longitudinal ornament. The transverse ornament of the tube consists of distinct annulation,

there being about 12–16 annulae per mm. Most tubes are discrete, but in some cases two or more tubes cross. The tube wall appears to be very thin, and there is no evidence of internal septae, pseudopunctae or punctae. Paired crescentic structures are preserved at the open end of the tube in some specimens.

**Host–symbiont association**. All specimens of *Vermilituus gregarius* are associated with vetulicolians, a group of extinct animals of disputed phylogenetic affinity that possessed a convex anterior part with frontal and lateral openings, articulating with a tail-like posterior extension (Figs. 2–6; Supplementary Figs. 1 and 2; for a summary of vetulicolians see ref. [6]). The soft anatomy of these animals is largely unknown, but the anterior part of *Vetulicola* has been hypothesised to comprise a pharynx with gill-like structures that flexed by means of horizontal and longitudinal muscle fibres attached to four flexible plates covered by a thin outer membrane (see below).

*Vermilituus gregarius* occurs in four specimens of *Vetulicola cuneata* from the Chengjiang region (Figs. 2, 4, and 5), plus six specimens of *Vetulicola rectangulata* from the Haikou region (Fig. 3; Supplementary Figs. 1 and 2; Supplementary data file). Overall, at least 400 specimens of *V. rectangulata* and 80 specimens of *V. cuneata* have been collected from the Chengjiang biota (YKLP and CJHMD collections), meaning that *Vermilituus gregarius* is a rare associate of vetulicolians. Vetulicolian fossils occur as composite moulds where the rock splits through the specimen, each part containing components of both the external and internal surfaces (Supplementary Fig. 3). For the anterior part of *Vetulicola*, we interpret *Vermilituus gregarius* as occupying the space between the interior of the exoskeleton, and the convex surface that appears to demarcate the position of the internal anatomy (Figs. 2a–c, 3a, b, g, 4a, b; Supplementary Figs. 1b–e, 2a, b).

The number of *Vermilituus gregarius* per associated vetulicolian is variable, ranging from a single tube to 88 individuals (Supplementary data file), and in some cases, *V. gregarius* occurs in local aggregates of up to 25 individuals, for example in YKLP 13075 (Fig. 3c, e, f). In most cases where *V. gregarius* aggregates, the individuals are discrete, but occasionally some overlap. The overall size of *V. gregarius* is from 0.8 to 7.2 mm in length, with maximum diameter ranging from 0.4 to 1 mm (proximal width is circa 0.2 mm). Average tube length varies within individual *Vetulicola* specimens (Supplementary data file), by a minimum of 1.6 mm (specimens associated with CJHMD 00031) to a maximum of 6.4 mm (specimens associated with YKLP 13073).

Rather than representing post-mortem assemblages, or the result of *Vermilituus* scavenging or colonising vetulicolian carcasses, all evidence suggests that *Vermilituus* attached to the body surfaces of living vetulicolians. All infested vetulicolians are preserved within 'event beds'[5]. This means that they were rapidly buried by sediment, and therefore post-mortem colonisation at the seabed is highly unlikely. Tubes of *Vermilituus gregarius* occur almost exclusively inside the vetulicolians, rather than the external body surface, and preferentially within the anterior part (Figs. 2–5; Supplementary Figs. 1 and 2). They are absent from other fossils preserved adjacent on the same slabs (Fig. 5a), and indeed have never been observed in other Chengjiang fossils in our investigations over the last three decades. Most specimens of *V. gregarius* occur in the anterior section of the vetulicolian body (*n* > 345) (Figs. 2–5, Supplementary Figs. 1 and 2), with just 4 specimens associated with the posterior section of the most-infested specimen in our collection (YKLP 13075, Fig. 3d). In this rare case, *V. gregarius* may have over-spilled onto the external surface of the animal or has been displaced post-mortem. Among those in the anterior part, most are located in the convex area

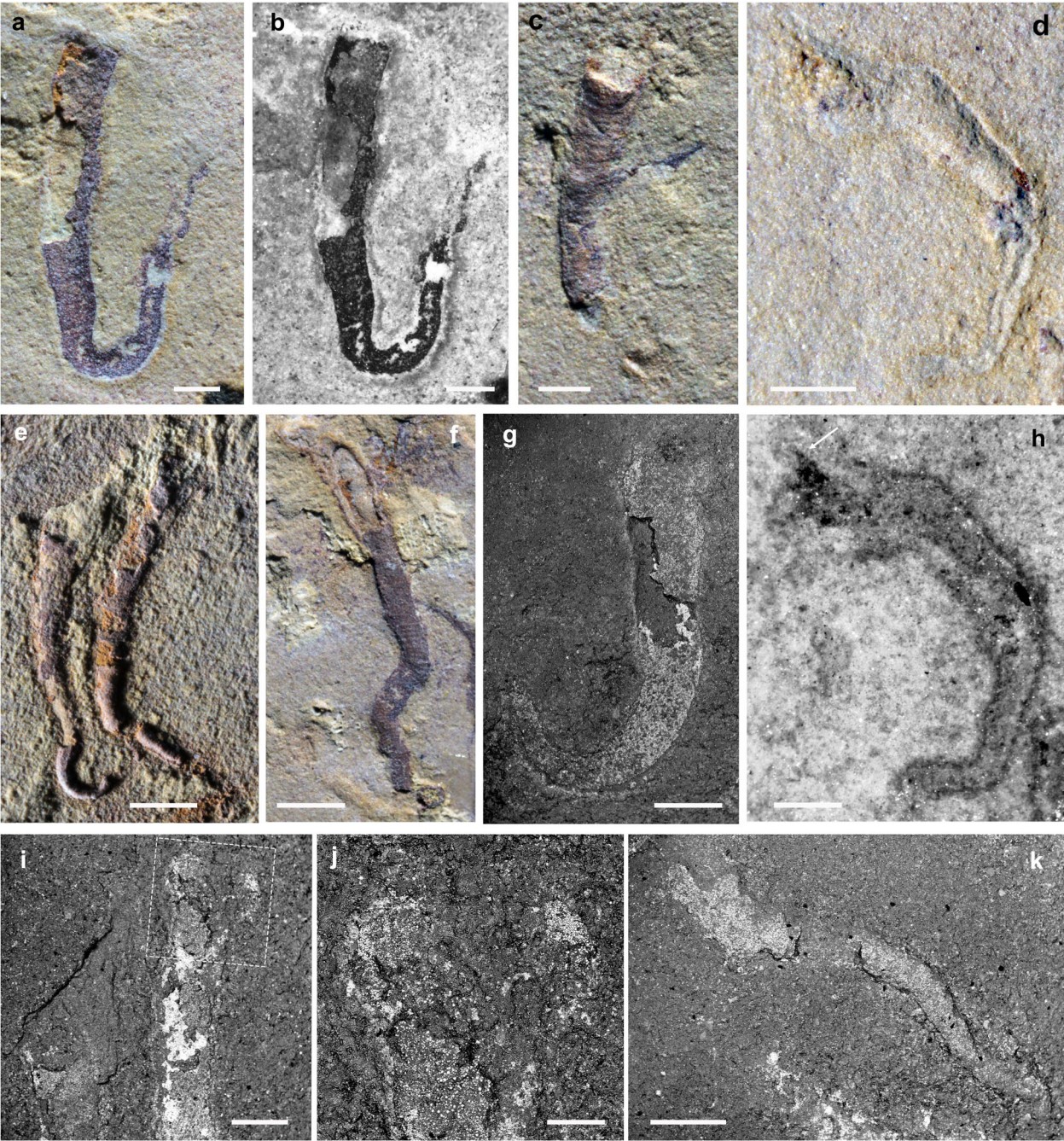

**Fig. 1 Different styles of preservation and morphology of _Vermilituus gregarius_. a, b** Holotype, YKLP 13079a, flattened specimen showing U-shape morphology, under cross-polarised light (**a**) and fluorescence light (**b**). **c** Paratype YKLP 13084, partial 3D with well-preserved annulation, J-shape morphology. **d, h** YKLP 13086 under direct light (**d**) and fluorescence light (**h**), white arrow shows possible soft tissues. **e** Paratypes YKLP 13082 and 13083, preserved in 3D with annulation visible proximally: sinusoidal shape and J-shape morphology, respectively (the latter is broken distally and shows sediment fill). **f** Paratype YKLP 13085, partial 3D with well-preserved annulation, sinusoidal morphology. **g, i–k** Scanning electron microscopy images. **g** YKLP 13087 with J-shape morphology. **i, j** YKLP 13088, boxed area in "i" shows possible paired soft tissues at the termination, magnified in "j". **k** YKLP 13089, with possible paired soft tissues at the terminal end. Scale bars: **a–d, g–i, k**, 500 µm; **e, f**, 1 mm; **j**, 200 µm.

between the central groove and the fin-like margins, with some concentrations often in the anterodorsal region (Fig. 3a, b). Only a few tubes of _V. gregarius_ occur along the margins of _Vetulicola_. In one case, at least 10 U-shaped tubes grow with a posterior orientation in _Vetulicola_ YKLP 10906 (Supplementary Fig. 1e). In _Vetulicola_ YKLP 13075, there is a clear association of 24 _V. gregarius_ with the central groove (Fig. 3f), each having a distinctive orientation with the narrow end of the tube pointing towards the groove.

The consistent occurrence of _Vermilituus gregarius_ inside the anterior section of vetulicolians, combined with the observed patterns of localisation and occasional preferred orientation (Supplementary data file), argues against a chance post-mortem association, or generalist epibiontic habit. In the latter scenarios, the posterior section should also be infested. Furthermore, _V. gregarius_ is absent from any other fossil organism in the Chengjiang biota, suggesting a highly specific relationship. The robust (possibly biomineralised) and curved tubes of _V. gregarius_

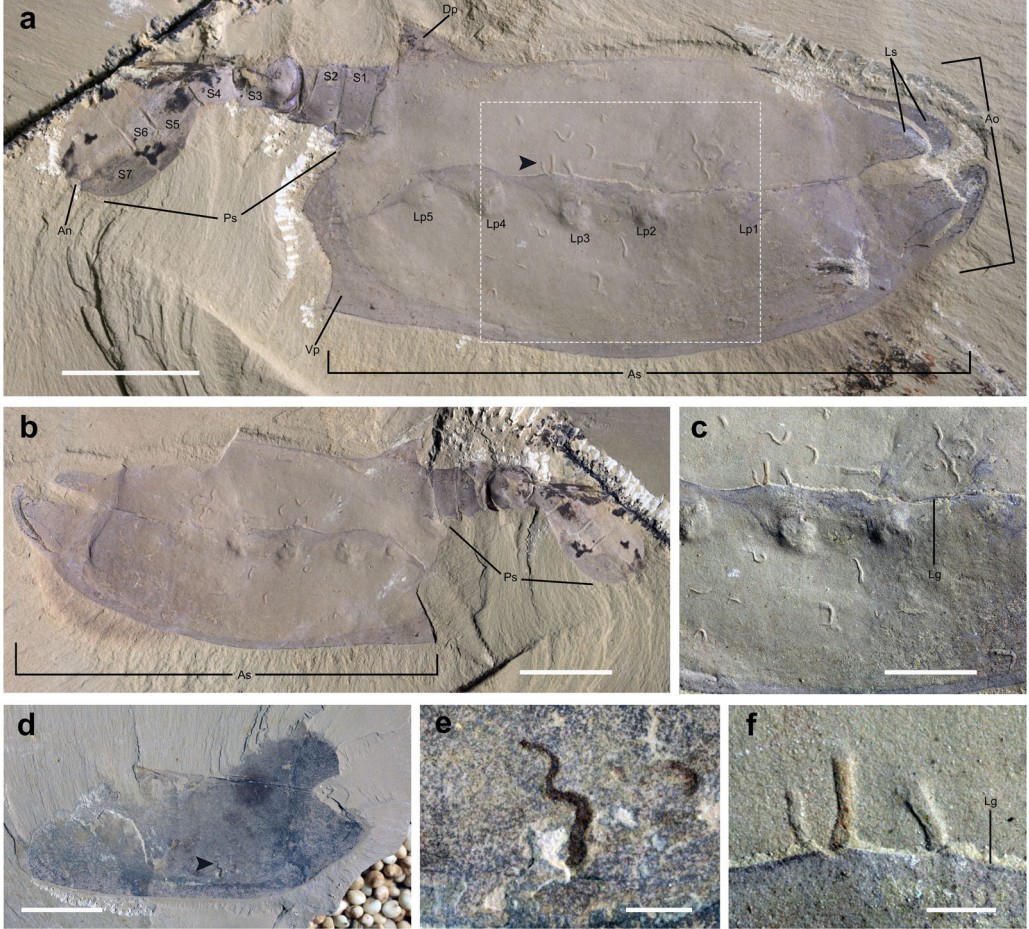

**Fig. 2 *Vetulicola cuneata* infested by *Vermilituus gregarius*. a** CJHMD 00031a, right view of the internal mould. Specimen infested with circa 20 *V. gregarius*. **b** CJHMD 00031b, left view of the internal mould. **c** Close-up of the area indicated in the box of image "**a**", showing concentration of *V. gregarius* specimens in the anterior section. **d, e** CJHMD 00032b, interior surface of the right side (dorsal to top) of the anterior section, and close-up of a sinusoidal *V. gregarius* tube. **f** Enlargement of arrowed area in image "**a**", showing concentration of three specimens along the central groove. An—anus, Ao—anterior opening, As—anterior section, Dp—posterodorsal projection, Lg—lateral groove, Lp—lateral pouch, Ls—lip-like structure, Ps—posterior section, S—segment, Vp—posteroventral projection (see ref. [2] for terminology). Scale bars: **a, b, d** 1 cm; **c** 5 mm; **e, f** 1 mm.

are consistent with a sessile, attached ecology, but not with a motile scavenger that might have fed on vetulicolians after death. The size range of *Vermilituus* on each specimen (Supplementary data file) suggests animals growing in situ for some time, rather than colonising carrion. In addition, the lack of evidence for decay and disarticulation of infested vetulicolians combined with their preservation in event beds supports an in vivo association. In this light, the observed patterns in size, number and distribution of *V. gregarius* tubes also shed light on vetulicolian biology and the ecological relationship between the taxa.

## Discussion

Based on their streamlined anterior section and tail-like posterior section (Fig. 6), vetulicolians have been interpreted as actively swimming animals[5], and some species may have been filter feeders[5,7–9], using the anterior section—which has been interpreted as a flexible pharynx—to gather food[9]. Vetulicolians have also been interpreted as nektobenthic consumers, feeding selectively at the sediment surface[5]. The host–symbiont relationships for *Vetulicola rectangulata* and *V. cuneata* described here support the idea of filter feeding by movement of water through the anterior section of *Vetulicola*[9], but the attachment of robust tubes of *V. gregarius* is not consistent with the hypothesis of active pumping of water facilitated via a flexible body wall that was

flexed by muscles. Our interpretation would support the presence of a space between the internal soft tissues and the inner surface of the exoskeleton of *Vetulicola* to accommodate endosymbionts, and also a degree of rigidity to the exoskeleton.

The earliest marine ecologies with benthic organisms constructing tubular structures are those of the terminal Neoproterozoic, and these likely included suspension-feeding animals[10,11]. None of these organisms constructed tubes that are similar to *V. gregarius* (Supplementary text), and none had a close symbiotic relationship encrusting a mobile host. Early Cambrian ecosystems, in contrast to those of the terminal Neoproterozoic, were populated by mobile, skeleton-bearing animals. These Cambrian ecosystems are known to preserve symbiotic associations, but they appear to have been dominated by epibiont suspension-feeding animals, most notably brachiopods, with a general absence of endosymbionts until the Ordovician[3]. Therefore, the endosymbiotic relationship between vetulicolians and *V. gregarius* adds a new dimension to Cambrian ecologies, with an unsuspected intensification of interactions early in the Phanerozoic.

*Vermilituus gregarius* tubes do not appear to represent the actions of animals scavenging on the *Vetulicola* carcass postmortem, as encrusting tube-bearing animals are sedentary. Postmortem colonisation of *Vetulicola* may be considered, as the

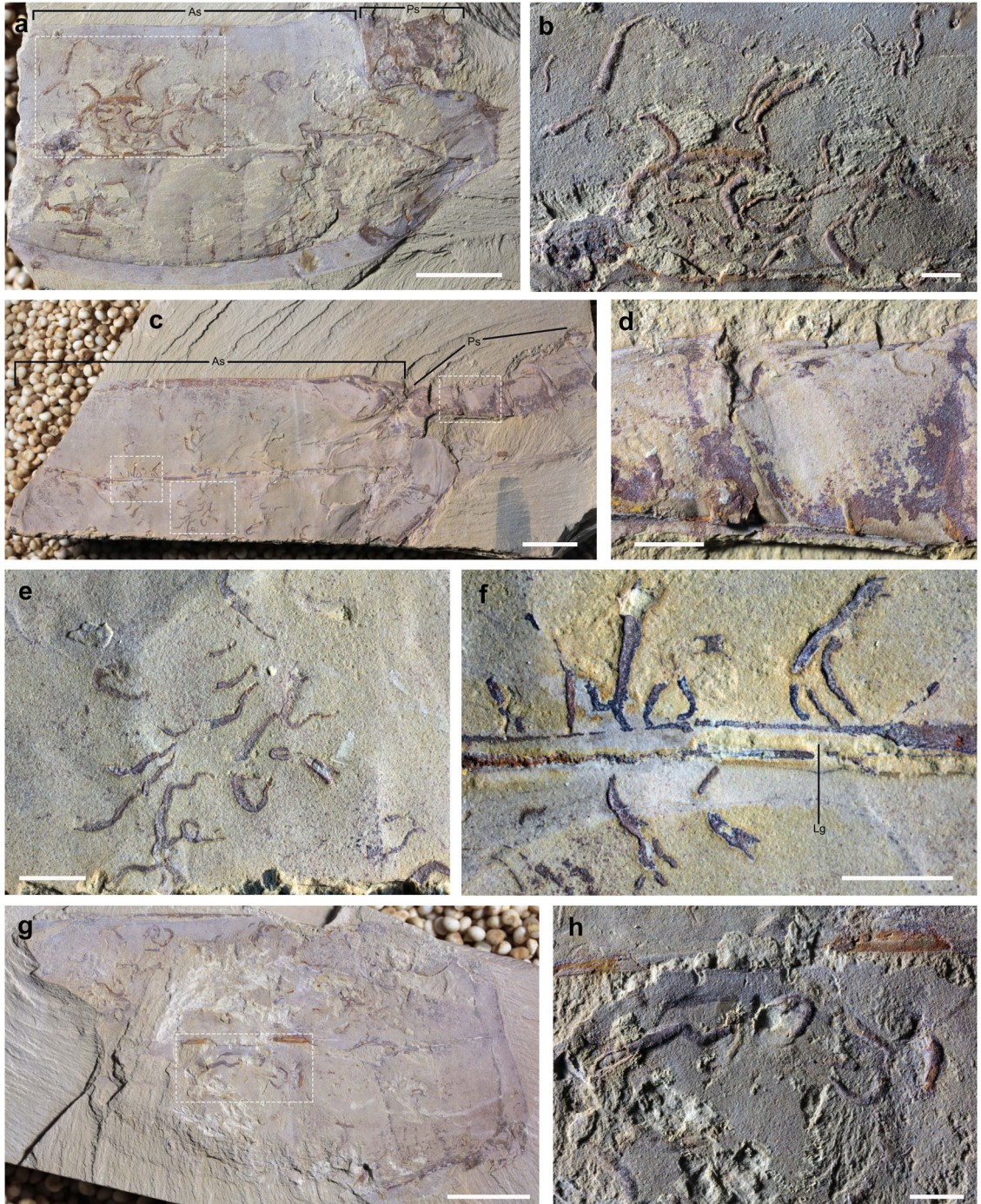

**Fig. 3 *Vetulicola rectangulata* infested by *Vermilituus gregarius*. a**, **b** YKLP 13073, left view of the internal mould of the anterior section (incomplete) and part of the posterior section. Specimen infested with circa 46 *V. gregarius*, with one aggregate toward the anterodorsal area (seen in "**b**") comprising 29 specimens. **c–f** YKLP 13075, left view of the internal mould of the anterior section, and composite mould of the posterior section infested with 88 *V. gregarius*, including three aggregates of between 10 and 25 specimens (e.g., seen in "**e**"), and concentration of 24 specimens along the central groove (close-up in "**f**"): note that these are oriented with the narrow end associated with the groove. This specimen also shows four specimens in the tail ("**d**"). **g**, **h** YKLP 13074, right view of the internal mould of the anterior section infested with about 52 *V. gregarius* that form aggregates of between 5 and 14, including those that preserve annulation ("**h**"). Scale bars: **a**, **c**, **g**, 1 mm; **b**, **d**, **e**, **f**, **h** 2 mm.

interior of the anterior section may have provided a protective habitat from predators and a stable surface for encrustation. However, if *V. gregarius* represents a filter-feeding animal, this would only facilitate food collection if the vetulicolian carcass was drifting through the water column for a protracted period after death. In addition, the vetulicolian hosts show no evidence of decay and/or disarticulation, which would be expected if *V.*

*gregarius* were simply exploiting dead and rotting carcasses for food, shelter or a substrate to settle on, whether in the water column or on the seafloor. The seafloor scenario is also unlikely because those vetulicolians that are infested were rapidly buried in 'event beds'[5].

Accepting a motile nektonic mode of life for vetulicolans, it follows that *Vermilituus gregarius* most likely infested the surfaces

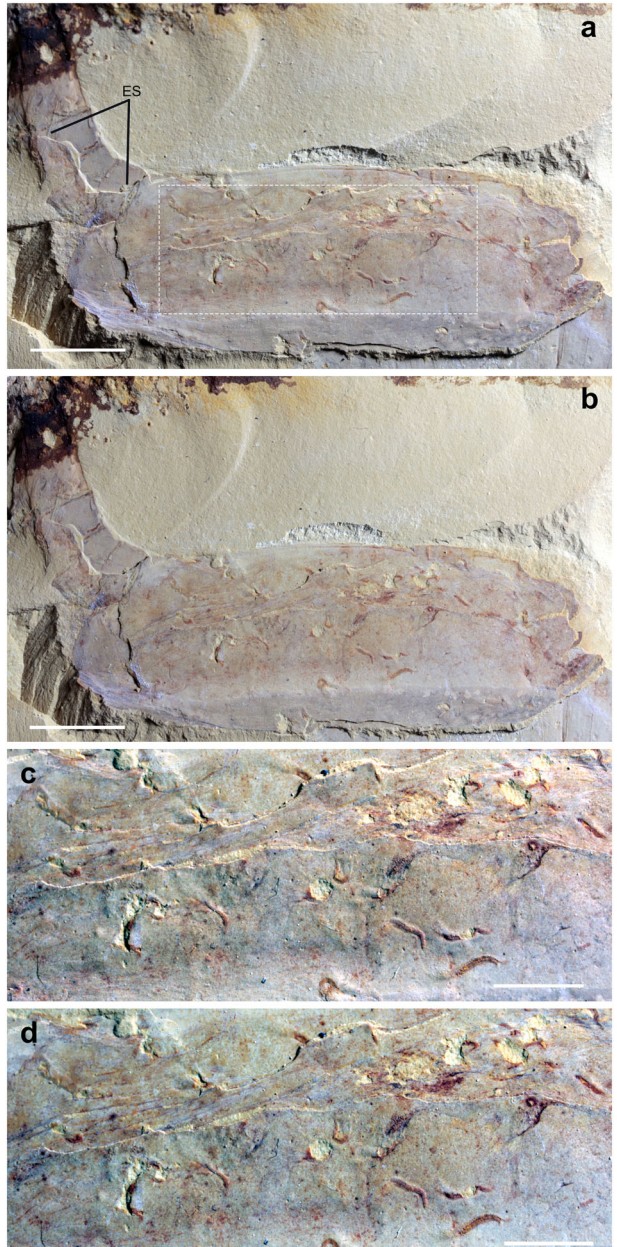

**Fig. 4 _Vetulicola cuneata_, CJHMD 00033 showing taphonomic relationships with _Vermilituus gregarius_.** Stereo images have a tilt of 20°, to emphasise that both the worms and the _Vetulicola_ are 3-dimensional. **a**, **b** Lateral view (stereo pair) of the whole specimen and **c**, **d** close-up of the anterior section (stereo pair), respectively. The specimen is a composite mould, with the external surface (ES) evident only in part of the posterior section, while most of the fossil shows an interior surface (see also Supplementary Fig. 3). Scale bars: **a**, **b** 1 cm; **c**, **d** 5 mm.

_Vermilituus gregarius_ is of uncertain taxonomic affinity (Supplementary text), but the paired, crescent-shaped structures seen protruding at the terminal end of some specimens (Fig. 1h–k) may be remnants of a feeding apparatus that might have been tentacular arms or a lophophore. The overall morphology of _V. gregarius_ is convergent with many living and fossil tubular annelids and possible lophophorates with a filter-feeding mode of life[12]. The aggregation of many _V. gregarius_ within the anterior section of the vetulicolians (Figs. 2–5, Supplementary Figs. 1 and 2) may suggest organisms utilising water currents (cf. oriented cornulitids on Ordovician brachiopods[13]) generated as part of the host's feeding mechanism or by the host's swimming direction (Supplementary Fig. 4). Patterns of aggregation and orientation occur locally, for example, the concentration of _V. gregarius_ along the central groove in YKLP 13075 (Fig. 3f) and CJHMD 00031 (Fig. 2c, f), which may have been associated with water expulsion[9]. But the absence of a consistent pattern of orientation for _V. gregarius_ may suggest complexity to the internal soft anatomy of vetulicolians, complex patterns of water circulation or that _V. gregarius_ adopted a different mechanism of acquiring food. It is notable that _V. gregarius_ has not been recorded from any other exoskeleton-bearing motile Chengjiang animal, including the many bivalved arthropods that present potential external and internal spaces for colonisation. Again, this suggests a high degree of specificity between host and symbiont.

Aggregative behaviour is a symbiotic strategy used by many tube-bearing animals[14]. The symbiotic association of _V. gregarius_ with vetulicolians is reminiscent of some fossil associations of serpulids that are interpreted to be commensal or mutualistic, for example, in corals and foraminifera[15,16]. The relationship between vetulicolians and _V. gregarius_ may have been commensal, the host providing a safe domicile, and the means of transport and distribution. Nevertheless, that four _Vetulicola_ each carry over 45 _V. gregarius_ suggests that infestation might have been deleterious to the survival of the host in some cases (Fig. 3a, b, g, h), and that _V. gregarius_ could therefore be parasitic. Competition for resources is particularly problematic if the host and parasite have similar preferences and clearance rates of consumed particles[17]. It is also possible that _V. gregarius_ would compete for oxygen, if the majority were residing alongside the host's respiratory apparatus. In addition, the tube-dwelling life mode of _V. gregarius_ suggests a more complex relationship than simple parasitism, as endoparasites do not typically require protective structures beyond those provided by the host.

The presence of even a small number of _V. gregarius_ tubes might have had a significant fouling effect on surfaces exposed to water flow (see ref. [9]). Parasitic fouling of otherwise regular structures would increase their surface roughness and therefore the hydrodynamic drag of fluid moving through the body cavity[18]. Growing populations of _V. gregarius_ within the host organism might not only impede fluid through-flow, but the additional weight of >45 individuals could have reduced the swimming efficiency of the host organism and its ability to feed (e.g., see Fig. 3a, b, g, h).

Infestation of the interior area of the anterior section in _V. rectangulata_ may have been encouraged by the wide anterior gape in its exoskeleton. _Vetulicola cuneata_ also possesses extended lip-like structures at its anterior opening[19], allowing larval _V. gregarius_ to enter, whilst other vetulicolians such as _Didazoon haoae_ and _Pomatrum ventralis_—which have no reported infestation—have narrow anterior openings that may have helped prevent infestation[20–22].

Together with the priapulid worms _Mafangscolex sinensis_ and _Cricocosmia jinningensis_, which are infested by aggregates of the small worm-like _Inquicus fellatus_[23], _Vetulicola rectangulata_ and _V. cuneata_ represent two of four species with host-specific

of _Vetulicola_ from the water column, presumably as gregarious larvae that settled mainly within the anterior section of the host rather than on its surface. This is suggested by the similar fidelity of preservation of both tubes and hosts, suggesting co-habitation (Supplementary data file). Although individual aggregations of _V. gregarius_ often contain specimens of a similar size, the overall size range in each vetulicolian suggests that larvae landed over a protracted period (Supplementary data file), and thus grew over some time. Aggregations do not appear to be the product of asexual budding, as in most cases the individual tubes are discrete (Figs. 1–5, Supplementary Figs. 1 and 2).

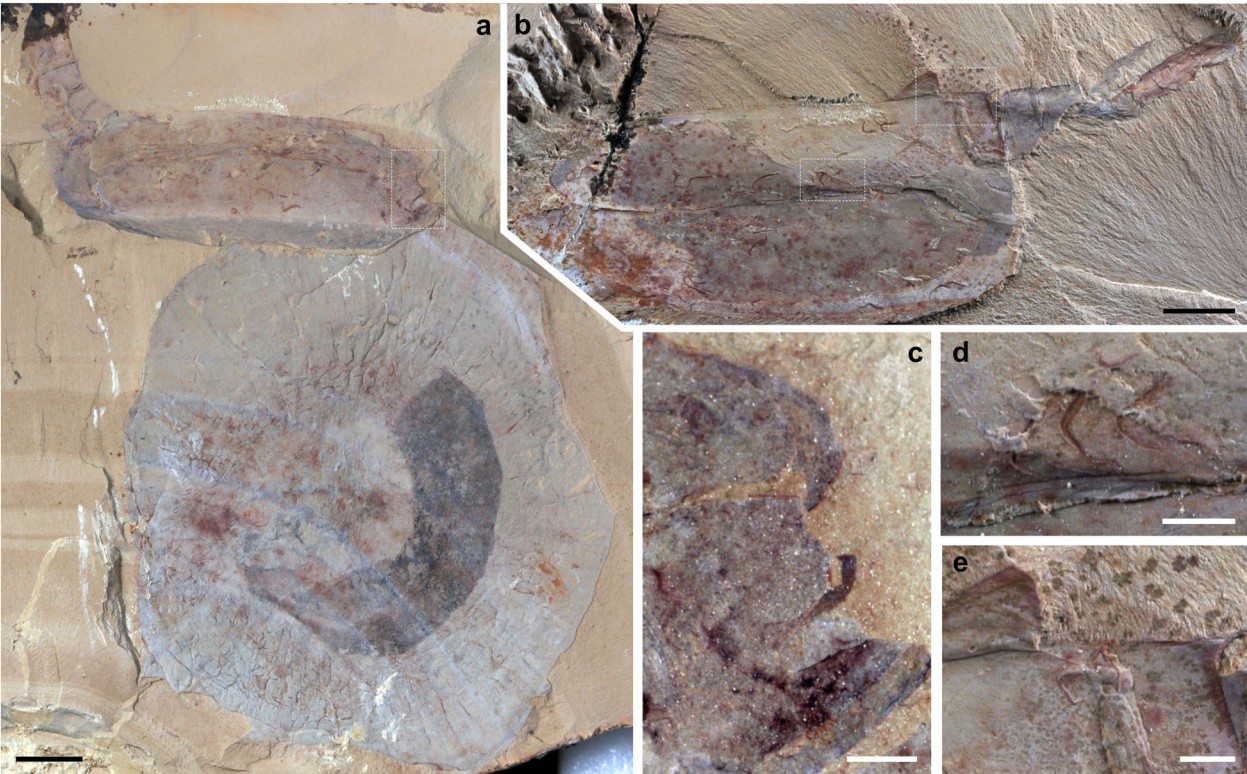

**Fig. 5 Host specificity of _Vermilituus gregarius_ with _Vetulicola cuneata_. a** CJHMD 00033, _Vetulicola cuneata_ preserved on rock slab with the fossil _Eldonia_. _Vetulicola_ infested with circa 17 _V. gregarius_. Note that _Eldonia_ was not infested. **b** CJHMD 00034, _Vetulicola_ infested with circa 34 _V. gregarius_. **c** Close-up of the area indicated in the box of image "a", showing one _V. gregarius_ specimen near the anterior opening. **d, e** Close-up of the area indicated in the box of image "b", showing concentration of three specimens along the central groove and one specimen at the position of the junction between the anterior and posterior section. Scale bars: **a, b** 1 cm; **c–e** 2 mm.

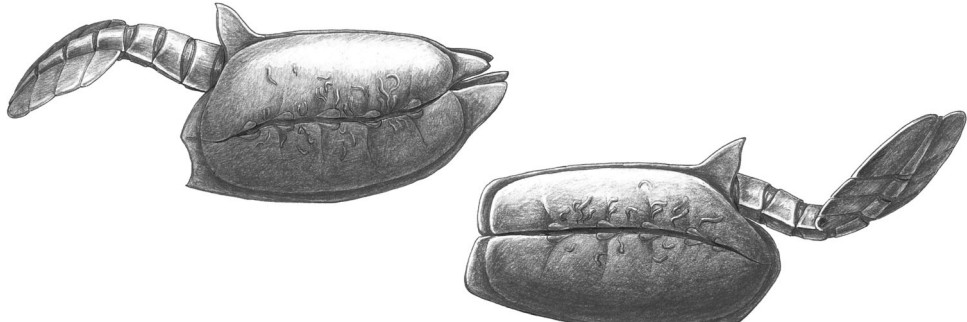

**Fig. 6 Reconstruction of _Vetulicola cuneata_ (left) and _V. rectangulata_ (right) in life.** Infestation by _Vermilituus gregarius_ is below the surface of the anterior section, that is, within the exoskeleton. Reconstructions are based on specimens about 6-cm long.

infestation documented in the Chengjiang biota. Of eight other vetulicolid species in the biota[6], none are infested. Infestation rates in _Vetulicola_ (10 individuals infested from circa 480 specimens overall) are low, representing a rate of about 2%, but this is within the range of, for example, modern polychaete infestations[24]. All infested _V. rectangulata_ are from three localities in the Haikou region, whilst the infested specimens of _V. cuneata_ are from Chengjiang County (Supplementary data file). Thus, _V. gregarius_ had a wide range across the marine basin of the Chengjiang biota[5]. The presence of _V. gregarius_ within the exoskeletons of two host _Vetulicola_ species suggests a propensity for host shift that is also demonstrated in worm infestations of Chengjiang priapulids[24].

Limited and conflicting evidence of the anatomy of vetulicolians means that their wider taxonomic affinities with other bilaterians are poorly understood. Thus, vetulicolians have been proposed to have affinities with arthropods[19,25], kinorhynchs[8], stem-group deuterostomes[9,22] and chordates[26,27]. Whatever their relationships to other bilaterian groups[28], the presence of a second group of animals in the Chengjiang biota with host-specific infestation and host shift (see also ref. [23]) indicates that such ecologies were likely widespread in Cambrian ecosystems. Early Cambrian ecosystems preserve rare symbiotic associations[4,29], but the fossil record shows a bias towards epibiont suspension-feeding animals, most notably brachiopods, with a general absence of endosymbionts until the Ordovician[3]: this is suggested to be due to reduced predation pressure in Cambrian ecosystems. Contrary to this, our data show that endosymbiotic and fouling relationships between species formed a component of the diversification of animal-rich ecosystems at the beginning of the

Phanerozoic Eon. Given that fouling strategies may substantially change the viability of the host, for example, through physical damage, reduced food supply and mechanical interference (ref. 17 and references therein), fouling may have been a substantial, and hitherto unrecognised driver of evolution in Earth's earliest animal-rich marine ecosystems.

## Methods

**Materials**. The fossils were collected from outcrops of the Yu'anshan Member, Chiungchussu Formation, *Eoredlichia-Wutingaspis* trilobite Biozone, Cambrian Series 2, Stage 3, Yunnan Province, China (Supplementary data file). All specimens are from 'event beds'[5]. Some specimens were prepared mechanically with needles under a stereomicroscope.

**Photography**. The digital images of the specimens were captured with a Canon EOS 5D SR camera mounted with Caron MP-E 65 mm (1–5×) or Canon EF 100-mm macro lenses under cross-polarised light, and were processed in Adobe Photoshop CC 2018. All measurements were processed with ImageJ version 1.49.

**Scanning electron microscope and energy-dispersive X-ray spectroscopy**. Scanning electron microscope (SEM) images (Fig. 1g, i–k) were obtained using a Hitachi S3600N at 10-kV accelerating voltage and 30-Pa chamber pressure. Energy-dispersive X-ray spectroscopy (EDX) was performed using an Oxford INCA 350 EDX at 15-kV accelerating voltage and 30-Pa chamber pressure. No evidence for phosphorus, calcium or carbon was identified in our analysis of the tubular shells, which are largely preserved by iron oxide after pyrite, a style of preservation that is widespread in Chengjiang fossils[30].

**Nomenclatural acts**. This published work and the nomenclatural act it contains has been registered in ZooBank, the proposed online registration system for the International Code of Zoological Nomenclature (ICZN). The ZooBank LSIDs (Life Science Identifiers) can be resolved and the associated information viewed through any standard web browser by appending the LSID to the prefix "http://zoobank. org/". The LSID for this publication is urn:lsid:zoobank.org:pub:4C4CD10E-3C65-4900-8CE9-1A1DC35E2B33.

**Reporting summary**. Further information on research design is available in the Nature Research Reporting Summary linked to this article.

## Data availability

Specimens YKLP 13073–13077, YKLP 13079, YKLP 13082–13089 and YKLP 10906 are deposited at the Yunnan Key Laboratory for Palaeobiology, Yunnan University, Kunming, China. Specimens CJHMD 00031–00034 are housed at the Chengjiang Fossil Museum of the Management Committee of the Chengjiang World Heritage Fossil Site, China.

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

## Acknowledgements

This study is supported by the National Natural Science Foundation of China (No. 41572015), State Key Laboratory of Palaeobiology and Stratigraphy (Nanjing Institute of Geology and Palaeontology, CAS) (No. 143101), Key Project of Yunnan Applied Basic Research (2017FA020), and Yunnan Provincial Research Grant (Nos. 2017FA020 and 2019Y0017). MW thanks the Leverhulme Trust for a Research Fellowship (RF-2018-275 \4). The reconstructions of Vetulicola in Fig. 6 were drawn by Xiaodong Wang.

## Author contributions

Y.L., P.C., X.H., F.W. and J.G. collected the material and, with M.W., devised the study. Y.L., P.C. and Y.Z. conducted the photography. All authors contributed to the interpretations of the fossils. T.H., M.W., S.G. and T.F. undertook SEM and EDX analyses. All authors discussed the data. Y.L. devised the figures. M.W. and Y.L. wrote the draft of the text with contributions from all the authors.

## Competing interests

The authors declare no competing interests.
