## [Peer Review File · Communications Biology]

Reviewers' comments:

Reviewer #1 (Remarks to the Author):

This MS presents interesting fossil evidence but there is a lack of discussion concerning possible modern analogues. This is the major weakness of the MS. The hypothesis of endo-parasitism needs to be clarified. Two hypotheses should have been tested 1) ectoparasitism 2) endoparasitism. There is no basic information on the internal anatomy of vetulicolians (where parasites are supposed to be attached). This MS needs substantial revision.

Reviewer #2 (Remarks to the Author):

The article "Endosymbiotic fouling of an early Cambrian nektonic animal", although I don't fully agree with the author's interpretation, they represent the relationship between multiple animals during the Cambrian explosion. If the author can clarify the relevant questions, it will help us to understand the complex ecological interactions in deep early.

The authors described a new animal fossil Vermilituus associated with Vetulicola. I agree with that two animal represent symbiotic relationship. Whereas, the description that the Vermilituus attached to the inner body surfaces of living vetulicolians need more supportive evidences. It's more reasonable that the Vermilituus attached on the outer surface of Vetulicola, instead the inner surface.

All the specimens in the figures described as internal mold of Vetulicola. If the small tubes attached to the inner surface, the question are:

- 1) The small tube preserved as convex mold on the Vetulicola, Is the small tube attach to the inner surface of carapace or soft tissues of Vetulicola?
- 2) In the article the authors declare "the robust tubular forms of the inhabitants suggest a sedentary surface-encrusting habit rather than invasion of the soft-tissues", where is the "carapace" of the Vetulicola?
- 3) I think the upmost layer most probably represents the carapace of specimens. If the tube attached to the inner surface of carapace, the tubes should exhibit attachment structures, rather than fine annulations.
- 4) Fig 5c, a small tube seems extend out the soft body, how to explain?
- 5) Vermilituus have the robust tube and distinct annulations. If Vermilituus is parasitic or endosymbiotic in Vetulicola, what is the function of these ornamentation and hard tube? And the coarse tube should stimulate the soft body, causing swellings and galls on their hosts, but cannot observed in the figures.

That the small tube attached to the internal body surfaces is the key evidence to conclude endosymbiosis and parasitism relationship. Can the authors provide more evidences? such as:

- 1) The distinct inner "carapace" surface bearing the small tubes or other structure including scars or pits, for example Fig1 in Huntley & Baets 2015(Huntley, J.W., De Baets, K., 2015. Chapter Five - Trace Fossil Evidence of Trematode—Bivalve Parasite—Host Interactions in Deep Time, in: De Baets, K., Littlewood, D.T.J. (Eds.), Advances in Parasitology. Academic Press, pp. 201-231)
- 2) Using undestroyed tomographic techniques construct the interior 3D morphology of fossil parasites inner the hosts, for example Siveter et al 2015(Siveter, David J, Briggs, Derek E.G., Siveter, Derek J., Sutton, Mark D., 2015. A 425-Million-Year-Old Silurian Pentastomid Parasitic on Ostracods. Current Biology 25, 1632-1637.)

The parasitism are difficult to diagnose from fossils. The lagerstätten Chengjiang Fauna have the best potential for the fossilization of parasites associated with their hosts. I hope the article can be published after providing more convincing evidences.

Some minor suggestions:

P4, L16-17: conical tubes having three general forms.....,

The tubes have various curve shape, the U-shape, J-shape, or complex sinusoid, these

three forms is not exact description. Additionally, U, J-shaped tube are easily confused with the trace fossil's terms.

P4, L20, Tubes increase in diameter very slowly, not exact. Can you give an expanding angle?

P4, L23, some cases two or more tubes cross, do you mean crosscut or overlap? same in P5, L15

P5 L16-20, provided data are inconstant with Sup Table1

Supplementary Figures and table,
Place each figure on a new page.
The table title is placed at the top of table.

Reviewer #3 (Remarks to the Author):

This is a great little paper, a pleasure to read. These biotic interactions are generally overlooked in Lagerstätten, so it is really nice to see this association be recognized and documented.

I do however, worry that the authors have overstretched their interpretations a little. At the moment things are quite subjective with little data presented. I agree that the tubes appear to be located on the internal surface of the vetulicolian specimens. Their presence is distinct and sometimes in such large numbers, which is very interesting in itself. The authors however, have followed this up with statements about preferred orientation and patterns of localization and used this evidence to support their claims against a chance post-mortem association or generalist epibiotic habit and even the life habit of the mysterious tube-dwelling organism. But very little evidence is actually presented to support this claim.

The statement regarding preferred orientation is based off a single vetulicolian specimen where 10 specimens of Vermilituus are possibly oriented towards the posterior. The orientation of each tube is not overly clear in the illustration and their orientation seems to vary quite a bit. When you look at the other specimens, there does not appear to be any preferred orientation. On first appearance, the tubes look to be going every which way. This is not to say that there couldn't be some preferred orientation, but no quantitative data is presented to support this.

The same can be said in regards to the patterns of localisation. The large majority of tubes are said to occur in the anterior section of the vetulicolian body, with only a few tubes found within the posterior section. This appears to be true, but considering that the anterior section of a vetulicolian represents the large majority of the organism (at least in terms of the surface area of the fossil specimens), I don't know if you can really call this a localized occurrence of the tubes. The authors go on to specify that the tubes are mostly located in the anterodorsal region, which is sometimes evident (like in Fig. 3a), but I wouldn't say it's consistent (in Fig. 3c there looks to be more tubes in the anteroventral region). In fact, in your table in the supplementary information, the term 'random' is used frequently to refer to the position of Vermilituus.

This preferred orientation, is frequently cited throughout the text and is also used to suggest that Vermilituus may have been utilizing the water currents generated by the vetulicolian's feeding or just by the vetulicolian's swimming direction. If this was true, and they were filter feeders (which is entirely plausible, however the supposed paired, crescent-shaped structures in Fig. 1 that may represent tentacles or a lophophore are not particularly clear and do not really strengthen the argument), I would expect that the tubes would be oriented towards the anterior opening, where the current would be entering the host. For example, the oriented cornulitids encrusted on the Ordovician brachiopods that are cited in the text, are oriented towards the commissure of the host, where the current produced by the brachiopod's lophophore would enter the valve. When I look at the specimens in Fig. 3, I cannot obviously see any preferred orientation, let alone

towards the anterior opening.

The curved, and sometimes sinusoidal nature of the tubes complicates this issue as well. If the tubular organism was filter feeding from the current produced by the host, why would they change their orientation of growth in such a manner? Regarding the tubes that are concentrated along the central groove, presumably the vetulicolian would have already filtered the nutrients out of the water before expulsion, so I'm not sure if this position would have been so advantageous for a filter feeding organism. That said, I'm not sure that this association of tube and central groove represents a significant relationship. It is only observed in two vetulicolian specimens and only a small percentage of the total number of tubes and it does not appear to be a consistent relationship.

To tell you the truth, my first impression was that this association represented infestation of the vetulicolian post-mortem. But I could be convinced either way, it is always difficult to definitely prove the true nature of these biotic interactions in the fossil record. Endosymbionts that live in a tube are very interesting though, are there any endosymbionts in extant taxa that live in a tube, chitinous or mineralized?

Overall, this is a very interesting paper and I think it will be of interest to not just palaeontologists, but also evolutionary biologists and ecologists. These interactions deserve publication, however I would just be looking for a bit more data and evidence to be presented to support the claims that are made throughout the paper. Since the orientation of the tubes is used to support many of the claims, maybe some simple rose diagrams or another suitable analysis to show if there is any preferred orientation would be beneficial. Regarding the localisation of the tubes, an analysis examining how many tubes are present in particular regions of the host specimen (at a smaller scale than anterior vs posterior), would be useful. There are enough individual tubes in the 10 vetulicolian specimens to produce a suitable dataset.

Also as a small editorial comment, in Supplementary Fig. 1. a caption is missing for 'e'. Also in the text, specimen YKLP 10906 is cited for the occurrence of 10 U-shaped tubes with a posterior orientation, but the figure caption for 'e' is missing and the arrows pointing to 10 tubes are in 'd' and specimen YKLP 13076.

Thanks for an enjoyable read and I also willing to review a revised version of the manuscript and feel free to contact me if questions arise.

Tim Topper

Dear Editor,

We thank the reviewers for their comments and suggestions. We note that reviewer 1 and 2 both have issues with the interpretations which stem from their being unconvinced that the tubes infest internal surfaces; they appear to require additional supporting evidence regarding the internal soft-part anatomy of *Vetulicola* to support our interpretation. We have added an additional supplementary figure to show how we infer inner and external surfaces, but note that little is known about the internal soft anatomy beyond that which we discuss – with reference to previous works – in the manuscript.

What have done in this manuscript is describe and illustrate carefully in multiple specimens (and using photographs and figurative drawings) the evidence that tubes attach to internal surfaces. We accept this is surprising (and that is what makes them interesting). Following we respond in detail to *all* the reviewers points. In some instances where there are requests from the reviewers that we believe are unlikely to yield a meaningful outcome, we have provided a clear explanation of our response.

Our responses are in blue text.

Reviewer #1 (Remarks to the Author):

This MS describes parasitic relationships between vetulicolans and unknown tubular organisms. It is the second example of such relationships in the early Cambrian ecosystem. This MS needs to be improved, especially by testing hypotheses more clearly (ecto versus endo-parasitism). It is unclear whether these parasites were attached to the external surface of the animal or to tissues or cuticle inside the body. Simplified diagrams showing the internal anatomy of vetulicolians (and the possible attachment sites of parasites) should be provided. It is not clear how these parasites could have infested the inner part of the posterior region. The main weakness of this MS is the lack of precise discussion concerning the affinities of these parasites in the light of possible modern analogues. There is a substantial literature on fouling and parasites in various animal groups.

For these main issues raised by the reviewer, we have made revisions and adjustments as detailed below.

Abstract

“Earliest fossil record ?”

What about Cong et al. about infestation in early Cambrian worms ? This key paper should be mentioned here.

Cong et al is cited in the text (several of us co-authored that paper too). The infestation reported by Cong et al. is from the Chengjiang Biota of the early Cambrian, the same deposit, so both of these are the earliest occurrences of these styles of symbiosis.

“Biofouling is a threat to animals”

Not always (see remarks below)

Host-symbiont association

We have modified to “biofouling is ubiquitous in coastal ecosystems”

Paragraph 3

What do you mean by “Tubes of *V. gregarious* occur exclusively inside the vetulicolans”. Do you mean that they are attached to the inner surface of the anterior body wall ?

First, I would recommend you to give more details about the anatomy of vetulicolians. Most readers have probably no idea concerning the anatomy of these weird animals. Please draw an idealized cross-section through a vetulicolan (add one figure) so that external and internal surfaces could be defined.

Since your specimens show no sign of decay/disarticulation (introduction paragraph) how is it possible that you could observe parasites attached to the inner surface of the animal ?

We show in multiple images that *Vermilituus* was attached to the inner surfaces of the anterior section of vetulicolians. Chengjiang fossils split across two slabs of rock - the “part and counterpart”, so we can observe many inner surfaces (see the new Suppl. Fig. 3). This is now made clearer in the figures too (extensive labelling) and the Table in the supplementary files. *Vermilituus* clearly occupies the space between the inner side of the skeleton and the non-skeletal soft tissues. *Vermilituus* does not occur on the surfaces of any of the other over 200 Chengjiang taxa.

There is no sign of disarticulation of the host and symbiont, and both have a similar state of preservation, showing the symbiosis occurred in life. The reviewer asks for a cross-section illustrating the internal anatomy of vetulicolians – unfortunately other than hints of possible musculature and a gut little of the internal anatomy of vetulicolians is known. However, there is no reason to assume that vetulicolians were ‘solidly’ filled with soft tissue in life; indeed, it is more likely that they had internal cavities, and indeed there must have been some internal space for fluid to flow (see the new Suppl Fig. 4).

The “composite moulds” need to be clarified, too. Perhaps, again, a diagram would be helpful here.

Vetulicolian fossils (and indeed all Chengjiang fossils) typically occur as composite moulds where the rock splits through the specimen, each part containing components of both the external and internal mould. We have already introduced this compound mode of preservation in the text, Supplementary Table 1, and in Supplementary Figure 2a we have pointed out the composite mould with external (ES) and internal surfaces (IS) on the figures. Please also see the new supplementary Fig. 3.

“Argue against a chance of postmortem association”

A more solid evidence for epibiont habit would be a vetulicolan showing parasites attached to either external side of its body. Is it the case?

We do not understand the reference to epibionts on the external surface and think this is a misconception. With the exception of 4 specimens in one heavily infested vetulicolan, all *Vermilituus* are attached to the inner surfaces. They are not attached to the external surface, as evident from Figures 2-5, and supplementary Figures 1 and 2.

Also please note, *Vermilituus* is not found on the surface of any of the over 200 other taxa in the Chengjiang biota, represented by 10s of thousands of specimens. It is not found on the outer or inner surfaces of the many bivalved arthropods in this deposit, even though such surfaces might resemble the functionality of the exoskeleton in *Vetulicola*. Instead this is a very specific host-symbiont relationship, as we argue.

Likely biomineralized

Why do you think these tubes are biomineralized. Perhaps elemental mapping would resolve this question ?

We have reworded to ‘possibly biomineralised’. The structures are mostly 3-D, whereas the vetulicolians are less convex. We think it is reasonable to suggest the original tubes were biomineralized, as Chengjiang fossils (like trilobites) that were mineralized are preserved in 3-D. We did indeed try elemental mapping, as is recorded in the manuscript (and referenced). All original biominerals in the Chengjiang biota are lost – as here, so elemental mapping does not reveal any additional information on shell composition.

Discussion

Please give more details about the circulation of water through the animal. Again you could easily make a simplified reconstruction. Circulation of water is crucial to the development of possible parasites.

The circulation of water through *Vetulicola* has been reconstructed by Ou et al. 2012 (as noted in the manuscript). Although our interpretation of the symbionts as internal would tend not to support their interpretation of a flexible pharynx, we do agree that the anterior part of *Vetulicola* likely functioned for food collection. Please also see the new Suppl.Fig. 4.

“The endosymbiotic relationship between vetulicolians and *V. gregarious*”

First you should made it clearer that these parasites lived inside the animal.

We show extensive photographic evidence that these animals are preserved *within* the anterior section of the vetulicolians.

“*Vermilituus gregarious* tubes do not appear....period after death”

Is not very clear to me. You should test two hypotheses: 1) the parasites colonized the carcasse after death. This colonization can be very fast (a few hours; e.g. protists inside ostracod carapaces) without any visible decay on the carapace. 2) in vivo infestation.

The protistan analogy inside ostracod carapaces doesn't work. The symbionts here are living in tubular skeletons, varying in size, and for each tube there is an increase in diameter from proximal to distal end, showing a period of protracted growth after colonizing the vetulicolian. This suggests the second hypothesis: *in vivo* infestation over a period of time, or, as we have entertained in the manuscript, that the vetulicolian was infested post-mortem whilst the carcass was still floating in the water column for a protracted period, providing continued access to food supply, though ultimately we reject this hypothesis, based on the lack of decay and disarticulation of the hosts.

“Water currents to facilitate respiration”

Here a diagram could explain the preferential location of parasites in areas where current flow is optimal (see my remarks above)

With the exception of one heavily infested vetulicolian where four *Vermilituus* are present on the possible exterior surface of the posterior part of the animal, all of the *Vermilituus* are concentrated in the anterior section, which is interpreted (e.g. Ou et al. 2012) to be the place where water currents flow (see Suppl. Fig. 4). One explanation for these four specimens in the posterior section is post-mortem shift, which we note in the manuscript, or overspill in a specimen that is very heavily infested (more than 80 symbionts).

In this section there is a lack of information concerning possible modern analogues (i.e. parasites living attached to the inner/outer surface of extant animals). I guess there are numerous references (including textbooks) that would provide accurate comparative data and illustration. Please add one paragraph and relevant illustration.

A key-reference is missing here: that of Weitschat who described large protists attached to the appendage of a Triassic myodocopid ostracod.

For *Vermilituus*, we provided a very detailed list of analogous animals in the supplementary file, most notably comparison with the serpulid worms. We make extensive comparison and discussion of other infesting tubular structures in the supplementary file. There is no suitable analogue for vetulicolians in modern animals, which is the difficulty of studying Cambrian enigmatic fossils. This is the case for both vetulicolians and *Vermilituus*. We do not consider the Weitschat paper a suitable analogue. Vetulicolians are not carapace-bearing arthropods, and *Vermilituus* is not a protistan.

“Carry over 50 *V. gregarious*”

I don't think 50 specimens of these tiny organism were deleterious to the survival of the host. They could have been deleterious if the parasites were attached to soft tissues such as gills. On what kind of biological substrate were they attached ? A very thin membrane, a cuticular layer. This point needs to be clarified and discussed.

We have modified the manuscript to suggest this only as a possibility. Four *Vetulicola* each carry over 45 *Vermiltuus*, 88 at the maximum. There is a possibility that these parasites attached to the internal functional structures of the host, and not just to the inner surface of the skeleton (the latter we can observe, the former is very likely but uncertain because of the absence of soft tissue preservation).

“Host’s respiratory apparatus”

Fist you have to explain how vetulicolans respired. The oxygen consumption of these tiny parasites was probably extremely low compared with that of a large animal like *Vetucolia*. Competition here does not seem to make much sense.

The anatomy of *Vetulicola* is uncertain, and no one knows how they respired. We take great care to indicate that in the manuscript. We are following other interpretations that suggest water flow in the anterior section (Ou et al 2012 and references therein). There is a place for inhalant and exhalent flow in vetulicolians (see suppl. Fig. 4). We have been circumspect and state the following: “It is also possible that *V. gregarius* would compete for oxygen, if the majority were residing alongside the host’s respiratory apparatus.” We would be worried stressing this further, we have just discussed the implications of our scenario if over 80 parasites were found inside of the vetulicolian anterior section. Near the lateral groove and pouches, the parasites are indeed possibly affecting the host’s respiratory apparatus, if we follow existing interpretations.

“Would increase their surface roughness”

Again such flexible soft-bodied parasites are unlikely to have disturbed water circulation within the body.

The parasites have tubular structures preserved with shell annulations and possible tentacles, which are not streamlined and would increase the surface roughness, so we don’t understand this question. For the internal surface this would affect the water circulation in the vetulicolian body.

“Reduce swimming efficiency”

I don’t believe this. The weight of these parasites was negligible compared with that of their host. Certainly no effect at all on the swimming efficiency. It would be different if the animal was colonized (externally) with brachiopod shells or barnacles.

I would recommend to delete these two statements and keep more space for modern analogues.

How can the reviewer know that the effect is negligible? These are fossils, and it is important to entertain this possibility. We do not know how vetulicolians propelled themselves. But it is clear that the posterior section is rather weak and slight in *Vetulicola* to operate as the only means of propulsion, and a water vascular system may have been involved in the anterior section. In this sense, our statement is reasonable based on the

evidence we have, and we choose to keep it.

As we are at pains to point out, there are no modern analogues for vetulicolians, but we have presented a long section on comparisons for *Vermilituus*, in the supplementary text file.

“The presence of four *V. gregarius*.... continued post-mortem;”

The fact that the posterior part is infested raises several questions. The posterior part of vetulicolians was most likely occupied by muscles and a gut tract. The hypothesis of parasites developing between the exoskeleton and the soft tissues is not very realistic. Below the cuticle you have epidermal cells, muscles and connective tissues. There is no empty space there that might be connected with the outside. How could the parasites possibly reach this part of the animal. The only possibility here is that the parasites were ectoparasites. This question needs clarification.

There is only one case where the parasites might be attached to the exterior surface (representing circa 1% of all specimens), and it is notable that this is the most infested specimen of all those we have examined. *Vermilituus* is not found on the external surface of any of the over 200 other species of the Chengjiang biota. This is a special and unique relationship, with colonization being essentially on the internal surfaces.

“Vetulicolians have proposed affinities with...kinorhynchs”

Hmm...Sorry but kinorhynchs have nothing to do with vetulicolians. They are scalidophorans (i.e. with proboscis and pharynx) ! You should delete this ref.

Here we are quoting other work and must be inclusive. Reviewers should not encourage authors to bias citations.

“Fouling strategies substantially change the viability of the host”

Not always. Take the whales which are covered with a variety of epibionts.

We state the following: “Given that fouling strategies substantially change the viability of the host, for example through physical damage, reduced food supply, and mechanical interference, biofouling may have been a substantial, and hitherto unrecognized driver of evolution in Earth’s earliest animal-rich marine ecosystems.” We hope this makes better sense when you read through the whole sentence.

We agree that not all fouling strategies damage the host, and so have added the word ‘may’.

Cong et al.’ paper about infestation in early Cambrian worms.

The parasites described in this MS should be compared with those found in early Cambrian worms (Cong’s et al.’s paper). Similar ? Different ? Distribution and cuticular damages in both species. Interesting discussions are missing here.

In the Cong et al. paper (that several of the authors here co-wrote), parasitic worms are intimately connected to the exterior integument of priapulid worms. Whereas we are examining encrusting worms in tubular skeletons on the internal surface of vetulicolians. There is no simple comparison to be made.

Reviewer #2 (Remarks to the Author):

The article “Endosymbiotic fouling of an early Cambrian nektonic animal”, although I don't fully agree with the author's interpretation, they represent the relationship between multiple animals during the Cambrian explosion. If the author can clarify the relevant questions, it will help us to understand the complex ecological interactions in deep early.

We have revised our interpretations based on the reviewers' comments and suggestions.

The authors described a new animal fossil *Vermilituus* associated with *Vetulicola*. I agree with that two animal represent symbiotic relationship. Whereas, the description that the *Vermilituus* attached to the inner body surfaces of living vetulicolians need more supportive evidences.

It's more reasonable that the *Vermilituus* attached on the outer surface of *Vetulicola*, instead the inner surface.

All the specimens in the figures described as internal mold of *Vetulicola*. If the small tubes attached to the inner surface, the question are:

1) The small tube preserved as convex mold on the *Vetulicola*, Is the small tube attach to the inner surface of carapace or soft tissues of *Vetulicola*?

We cannot be sure because little is known about the soft anatomy of the anterior section of vetulicolians. What is clear though, is that the worms occupy internal surfaces, as indicated by all known specimens.

2) In the article the authors declare “the robust tubular forms of the inhabitants suggest a sedentary surface-encrusting habit rather than invasion of the soft-tissues”, where is the “carapace” of the *Vetulicola*?

The ‘carapace’ (anterior section) is clearly labelled in our figures.

3) I think the upmost layer most probably represents the carapace of specimens. If the tube attached to the inner surface of carapace, the tubes should exhibit attachment structures, rather than fine annulations.

The surfaces we show in many figures are the internal surfaces of the ‘carapace’ and the internal mould (see explanatory suppl. Fig. 4). We have not seen *Vermilituus* on the external

surface, with one exception of four specimens in the posterior section. Annulation is present in encrusting organisms with a tubular skeleton, with which we make comparison in the supplementary text file.

4) Fig 5c, a small tube seems extend out the soft body, how to explain?

Yes, this is one outlier amongst many hundreds, and is likely displaced post-mortem. Note that *Vermilituus* is only ever associated with *Vetulichola* in the 10s of thousands of specimens of Chengjiang animals that have been collected over 3 decades.

5) *Vermilituus* have the robust tube and distinct annulations. If *Vermilituus* is parasitic or endosymbiotic in *Vetulichola*, what is the function of these ornamentation and hard tube? And the coarse tube should stimulate the soft body, causing swellings and galls on their hosts, but cannot observed in the figures.

The presence of galls and swellings is not observed, but the soft tissues of the vetulicholians are not preserved. We discuss at length comparable tubular organisms in the supplementary data.

That the small tube attached to the internal body surfaces is the key evidence to conclude endosymbiosis and parasitism relationship. Can the authors provide more evidences? such as:

1) The distinct inner “carapace” surface bearing the small tubes or other structure including scars or pits, for example Fig1 in Huntley & Baets 2015(Huntley, J.W., De Baets, K., 2015. Chapter Five - Trace Fossil Evidence of Trematode—Bivalve Parasite—Host Interactions in Deep Time, in: De Baets, K., Littlewood, D.T.J. (Eds.), *Advances in Parasitology*. Academic Press, pp. 201-231)

Our manuscript includes multiple images demonstrating that the tubular worms are on the internal surfaces of the vetulicholians (please also see the new suppl. Fig. 4). Please note above, that *Vermilituus* is not found on the external surface of any of the over 200 different taxa of the Chengjiang biota. This is a special relationship, with colonization on the internal surfaces.

2) Using undestroyed tomographic techniques construct the interior 3D morphology of fossil parasites inner the hosts, for example Siveter et al 2015(Siveter, David J, Briggs, Derek E.G., Siveter, Derek J., Sutton, Mark D., 2015. A 425-Million-Year-Old Silurian Pentastomid Parasitic on Ostracods. *Current Biology* 25, 1632-1637.)

CT techniques have been tried here, but there is insufficient mineralization to resolve discrete structures. The vetulicholians are only weakly 3-D and with no preservation of soft tissues, so the excellent tomographic techniques of our colleagues Siveter et al. unfortunately do not work here.

The parasitism are difficult to diagnose from fossils. The lagerstätten Chengjiang Fauna have the best potential for the fossilization of parasites associated with their hosts. I hope the article can be published after providing more convincing evidences.

We thank the reviewer for these comments.

Some minor suggestions:

P4, L16-17: conical tubes having three general forms.....,

The tubes have various curve shape, the U-shape, J-shape, or complex sinusoid, these three forms is not exact description. Additionally, U, J-shaped tube are easily confused with the trace fossil's terms.

We use these terms as descriptors of what we see and provide multiple visual images to confirm this.

P4, L20, Tubes increase in diameter very slowly, not exact. Can you give an expanding angle?

We give a detailed description of the tubes in the supplementary data.

P4, L23, some cases two or more tubes cross, do you mean crosscut or overlap? same in P5, L15

Overlap (see fig. 3b).

P5 L16-20, provided data are inconstant with Sup Table1

Double checked and corrected.

Supplementary Figures and table,

Place each figure on a new page.

The table title is placed at the top of table.

Corrected.

Reviewer #3 (Remarks to the Author):

This is a great little paper, a pleasure to read. These biotic interactions are generally overlooked in Lagerstätten, so it is really nice to see this association be recognized and documented.

We thank the reviewer for this comment. Documenting endosymbiosis will help evaluate the complexity of Cambrian marine ecosystems.

I do however, worry that the authors have overstretched their interpretations a little. At the

moment things are quite subjective with little data presented. I agree that the tubes appear to be located on the internal surface of the vetulicolian specimens. Their presence is distinct and sometimes in such large numbers, which is very interesting in itself. The authors however, have followed this up with statements about preferred orientation and patterns of localization and used this evidence to support their claims against a chance post-mortem association or generalist epibiotic habit and even the life habit of the mysterious tube-dwelling organism. But very little evidence is actually presented to support this claim.

We are very grateful that this reviewer recognizes that *Vermilituus* is on preserved on the inner surfaces of *Vetulicola*. We have made revisions according to the reviewer's comments (see below)

The statement regarding preferred orientation is based off a single vetulicolian specimen where 10 specimens of *Vermilituus* are possibly oriented towards the posterior. The orientation of each tube is not overly clear in the illustration and their orientation seems to vary quite a bit. When you look at the other specimens, there does not appear to be any preferred orientation. On first appearance, the tubes look to be going every which way. This is not to say that there couldn't be some preferred orientation, but no quantitative data is presented to support this.

The same can be said in regards to the patterns of localisation. The large majority of tubes are said to occur in the anterior section of the vetulicolian body, with only a few tubes found within the posterior section. This appears to be true, but considering that the anterior section of a vetulicolian represents the large majority of the organism (at least in terms of the surface area of the fossil specimens), I don't know if you can really call this a localized occurrence of the tubes. The authors go on to specify that the tubes are mostly located in the anterodorsal region, which is sometimes evident (like in Fig. 3a), but I wouldn't say it's consistent (in Fig. 3c there looks to be more tubes in the anteroventral region). In fact, in your table in the supplementary information, the term 'random is used frequently to refer to the position of *Vermilituus*.

The vast majority of the worm tubes appear in the anterior section of vetulicolians. We provide multiple images to show that the worm tubes are attached to the inner surface of the vetulicolians (see also the new suppl Fig. 4). The posterior section of vetulicolians was most likely occupied by muscles and a gut, with no cavity for respiratory apparatus or water flow. When the hosts were very highly infected, parasites may have extended to the external surface of the body, spreading from the anterior to the posterior section (the specimen in question is the most infested, with 88 *Vermilituus*).

Tubes are generally aggregated in the anterodorsal region (see Fig. 3a, g). However, to take on board this reviewer's comments we now write the following: "with some marked concentrations often in the anterodorsal region (Fig. 3a,g)".

This preferred orientation, is frequently cited throughout the text and is also used to suggest

that *Vermilituus* may have been utilizing the water currents generated by the vetulicolian's feeding or just by the vetulicolian's swimming direction. If this was true, and they were filter feeders (which is entirely plausible, however the supposed paired, crescent-shaped structures in Fig. 1 that may represent tentacles or a lophophore are not particularly clear and do not really strengthen the argument), I would expect that the tubes would be oriented towards the anterior opening, where the current would be entering the host. For example, the oriented cornulitids encrusted on the Ordovician brachiopods that are cited in the text, are oriented towards the commissure of the host, where the current produced by the brachiopod's lophophore would enter the valve. When I look at the specimens in Fig. 3, I cannot obviously see any preferred orientation, let alone towards the anterior opening.

This is a good point by the reviewer, and we have added statements to the text to qualify this. Nearly all of the specimens (about 99%) are concentrated in the anterior section of the vetulicolians, on the inner surfaces. The *Vermilituus* are not found on accompanying animals even where these are on the same lamination surface. There are places where there is a clear orientation and localization of the infesting worms, as we note in the Table (supplementary file) and text, and this can be observed in most of the specimens with strong infestation. Though we note (in the text and Table) that there are random orientations too. A problem here is that we do not know how the anterior section of vetulicolians functioned, and we do not know what the soft tissues looked like, or how they functioned. Our interpretation would not support the idea of a flexible pharynx and gills (Ou et al. 2012).

The curved, and sometimes sinusoidal nature of the tubes complicates this issue as well. If the tubular organism was filter feeding from the current produced by the host, why would they change their orientation of growth in such a manner? Regarding the tubes that are concentrated along the central groove, presumably the vetulicolian would have already filtered the nutrients out of the water before expulsion, so I'm not sure if this position would have been so advantageous for a filter feeding organism. That said, I'm not sure that this association of tube and central groove represents a significant relationship. It is only observed in two vetulicolian specimens and only a small percentage of the total number of tubes and it does not appear to be a consistent relationship.

To follow from the point above, we note that there are different orientations of *Vermilituus* within the vetulicolians, but nevertheless we cannot ignore their clear association with the central groove in two specimens, that in some specimens the worms cluster, and in some specimens there is a preferred orientation (see Table 1). We have qualified this in the text to be more circumspect, given that we do not fully understand the soft anatomy contained in the anterior section of these animals.

To tell you the truth, my first impression was that this association represented infestation of the vetulicolian post-mortem. But I could be convinced either way, it is always difficult to definitely prove the true nature of these biotic interactions in the fossil record. Endosymbionts that live in a tube are very interesting though, are there any endosymbionts

in extant taxa that live in a tube, chitinous or mineralized?

We have explained why this association couldn't be post-mortem (see above, reply to reviewer 1).

We provide multiple comparisons with other fossil organisms in the supplementary data.

Overall, this is a very interesting paper and I think it will be of interest to not just palaeontologists, but also evolutionary biologists and ecologists. These interactions deserve publication, however I would just be looking for a bit more data and evidence to be presented to support the claims that are made throughout the paper. Since the orientation of the tubes is used to support many of the claims, maybe some simple rose diagrams or another suitable analysis to show if there is any preferred orientation would be beneficial. Regarding the localisation of the tubes, an analysis examining how many tubes are present in particular regions of the host specimen (at a smaller scale than anterior vs posterior), would be useful. There are enough individual tubes in the 10 vetulicolian specimens to produce a suitable dataset.

We don't use the orientation of the tubes to support most of our claims, rather it is the fact that the tubes are on the inner surfaces. From the beginning (in Table 1) we noted that some orientations were indeed random. Nevertheless, most *Vermiltuus* are in the anterior section, and on the internal surfaces. And in many instances, they aggregate in certain parts, or concentrate on certain structures. We do need to point this out, whilst accepting that some are randomly orientated.

Also as a small editorial comment, in Supplementary Fig. 1. a caption is missing for 'e'. Also in the text, specimen YKLP 10906 is cited for the occurrence of 10 U-shaped tubes with a posterior orientation, but the figure caption for 'e' is missing and the arrows pointing to 10 tubes are in 'd' and specimen YKLP 13076.

Corrected.

Thanks for an enjoyable read and I also willing to review a revised version of the manuscript and feel free to contact me if questions arise.

Reviewers' comments:

Reviewer #1 (Remarks to the Author):

This is an interesting MS but there are still numerous points that need clarification. See my comments in attached file.

Reviewer #2 (Remarks to the Author):

I feel the authors have made some acceptable exposition of many points raised by the reviewers. I support the publication.

There are still some concerns worthy of careful consideration by the authors. The extraordinary Chengjiang fossils can preserve the soft tissues within the shales, which can be considered as three-dimensionally anatomical structure of organisms. As the authors illustrated in supplementary figure 3, if the "carapace" of the animal is concrete and solid shell, there is without any doubt that *Vermilituus* attached on the inner surface of *Vetulicola*. Whereas, the carapace of *Vetulicola* is preserved as membrane, or halo of disseminated soft tissues in the surrounding rock. Due to the tube of *Vermilituus* is robust and has pyritization to some extent, the small tube can be observed on multiple layers of sediment and fossil, for example, text fig 3b, 4b. The author claims that the appearance of the *vermilituus* tube on the inner surface of *Vetulicola*, which cannot be confirmed.

Figure 4, and Supplementary Figure 2, multi-panels should be labelled with a, b, c, d.

Reviewer #3 (Remarks to the Author):

Thanks once again for an enjoyable read. I'm glad to see this one back. I don't really have any further comments or suggestions to make on the manuscript. These interactions are so rare to capture in the fossil record, so it would be great to see this published and out there for people to read. Given the information at hand, I approve of the conservative stance taken regarding the type of symbiotic relationship we are seeing here, whether parasitic or commensal etc. I feel this is warranted given the mystery that surrounds both the symbiont and the host. Regardless, it is a very strange and interesting relationship, I don't know if I've heard of any endosymbiont that lives in a tube. But it's the Cambrian, so it would not surprise.

Just out of curiosity, what is the size range of *Vetulicola cuneata* and *V. rectangulata* documented from Chengjiang? And what are the sizes of the specimens that are infested with *Vermilituus*? You have listed the sizes of the tubes, but not of the host. Is it only specimens towards the larger end of the scale that are infested and do you see more tubes in the larger specimens, or is it just random? Just interested if there may be any relationship, even if the sample size is not large.

Thanks once again and I look forward to seeing this published.

Cheers,
Tim Topper

Please find our responses to the reviewers' comments in red text. We have made some modifications to the main body text based on these comments.

We welcome the comments of reviewers 2 and 3 that our manuscript can now be published. We have responded to the comments of all reviewers below, on a point-by-point basis.

Reviewer #1 (Remarks to the Author):

This is an interesting MS but there are still numerous points that need clarification. See my comments in attached file.

Abstract Line 24

“Our data show that symbiotic and fouling relationships between species already formed a component of the diversification of animal-rich ecosystems near the beginning of the Phanerozoic Eon”

Actually, it is Cong et al. (2017) who first demonstrated this kind of relationships. The present MS presents a second case, also from the Chengjiang Lagerstätte. Please change.

If we take the style of infestation as fouling, which we believe it is, our statement holds fully (the style of infestation reported in the Cong et al. paper –which we authored– was of a different nature).

Abstract Line 27

“Today, biofouling is ubiquitous in coastal ecosystems and this has been an economic nuisance for human-built structures and fisheries”

I don't think this sentence highlighting economic nuisance is relevant here. The real nuisance for marine environments comes from the antifouling agents used by humans ! Please delete this statement.

Here we are talking about human structures that are fouled, and therefore the comment we make is legitimate. We are not talking about damaging natural ecologies. We have reworded the abstract.

Line 33

Please add refs here (e.g. Vannier 2012, Zacaï et al.) on gut contents in *Ottoia* and *Sidneyia*

This is not the main focus of the paper, and the one review paper we cite is a good entry point to this literature, however we have added the Vannier reference as requested.

Line 44 Hosts show no sign of decay and disarticulation, host and symbiont are similarly preserved, and symbionts are not found anywhere else, indicating that the symbiosis was very specific and occurred in life

This statement is questionable. Many nekto-benthic animals die in the water column and reach the seafloor without being disarticulated. As soon as they die

(i.e. when autolysis starts) they are colonized by various organisms that attach to their membranes. Some are microscopic (e.g. protists) others are larger. I agree that vetulicolians were probably swimmers. Your hypothesize that symbiosis may have occurred in life in interesting but fails to convince me. There is an equally plausible option: that of freshly dead vetulicolians lying on the bottom and infested by Vermilituus. Please, consider these two hypotheses in your MS.

Firstly, we note that worms are never associated with heavily disarticulated vetulicolians. All of our specimens that are infested are from 'event beds', which means rapid sedimentation and burial. It is therefore most unlikely (probably impossible) that they could have attached at the seabed, unless they burrowed into the sediment as juveniles and found their way into the vetulicolian carcass. This is not a parsimonious explanation.

We did discuss the idea of the vetulicolians being attached post-mortem whilst the animal was floating. We wish to emphasise again that host and symbiont show very similar preservation, and that the infesting worms are only concentrated in the anterior section. If this was postmortem infestation before much decay (as the reviewer is suggesting), either whilst floating or on the seabed, then *all parts* of the vetulicolians, anterior and posterior, would be *equally* susceptible to worm attachment. But, in contrast what we record is a concentration of attachment by worms to the anterior of the vetulicolian – this preferential site of attachment is best explained by in life colonization. Indeed, preferential attachment of symbionts to specific anatomical parts of a host is often used as diagnostic of in life colonization. Furthermore, the rigid tube of the worms, whether biomineralized or not, would have taken considerable time to grow – probably much longer than the soft tissues of the vetulicolian survived decay or consumption by scavenging. We have strengthened the text to explain this.

Line 46

“the robust tubular forms of the inhabitants suggest a sedentary surface encrusting habit

rather than invasion of the soft-tissues”

You interpret Vermilituus as an encrusting animal comparable with extant serpulids (see below). However, I see no strong evidence that Vermilituus was an encrusting animal lying flat on the internal surface of vetulicolians. Instead, Vermilituus was most likely attached to the vetulicolian in a manner similar to a polyp (i.e. standing upright) as suggested by Fig. 3 f (see below; please note that all specimens seem to be aligned (red arrows)- this configuration is not found in encrusting organisms such as serpulids). It is an illusion to believe that Vermilituus was lying flat on the body surfaces of vetulicolians. This position simply results from compression. Please reconsider your interpretation or at least analyze alternative options to encrusting.

We believe this statement by the reviewer is a misrepresentation of our manuscript. Only at one point in the main manuscript do we state: “The symbiotic association of *V. gregarius* with vetulicolians is reminiscent of some fossil associations of serpulids that are interpreted to be commensal or mutualistic, for example in corals and foraminifera”. This is not suggesting that *Vermilituus* is comparable to serpulids, but simply points out some similarities. We go to great lengths in the supplementary file

to point all possible comparator organisms in the Cambrian.

And in the supplementary file we are even more circumspect: “*V. gregarius* superficially resembles the tubular structures of serpulid and sabellid polychaete annelids, though the earliest confirmed record of the former is Middle Triassic, and of the latter is Late Carboniferous⁷. The strongly 3-D morphology of *V. gregarius* tubes (Fig. 1c,e,f,k) suggests they may have been biomineralized, though no trace of shell material has been detected. SEM analysis reveals no evidence of wall microstructure, of internal structure, or whether the tube is open at the narrow end (Fig. 1g,i-k), and this combined with the age of the material means that comparison with serpulid annelids is speculative.”

And a point of correction to the reviewer, because serpulids can be aligned in ammonites and brachiopods. In the former they change orientation as they grow to maintain position relative to the aperture. In brachiopods they grow again towards the aperture (see: Seilacher, A., 1982. Ammonite shells as habitats in the Posidonia Shales of Holzmaden – floats or benthic islands? Neues Jahrbuch für Geologie und Paläontologie, Monatshefte 1982, 98-114, and <http://www.lymeregismuseum.co.uk/related-article/epifaunal-worm-tubes-lower-lias-ammonites-conclusions-references/>)

Yes, some worms are aligned but some are not. Suggesting they are all aligned is a misrepresentation of our figures and text. Where some worms are aligned this is intriguing. In the case pointed out by the reviewer they are associated with the putative exhalent region of the animal, and perhaps this is somehow related to water flow. Again, this alignment would not occur if the worms were attaching post-mortem (either in the water column or on the seabed). The big problem remains that the evidence suggests these animals as occupying the internal surfaces of the animal, and not attached to its outer surface. If they were attached like polyps, there would be issues of space. This polyp model also does not fit the shape of many of the tubes, which are often U-shaped and S-shaped, that is growing back towards their origin.

Line 48

“The number of inhabitants in some 48 specimens (>45) may have induced a negative effect on the host, the attaching endosymbionts partly obstructing water flow, i.e., biofouling of internal body surfaces”

This statement is also questionable. Vetulicolians are interpreted as organisms filtering water through paired lateral openings. None of your specimens show local concentrations of parasites near these openings. I am just wondering how then Vermiltuus could have obstructed water flow. The term “biofouling” is eye-catching although a bit excessive. Biofouling is supposed to cause structural or functional deficiencies. I can hardly believe that a few tens of these tiny organisms scattered over the internal surface of vetulicolans could have caused any severe damage to their host. Anyway they were probably growing in carcasses.

This comment that this is not biofouling seems to be a misreading of the data (we do not claim to have 48 vetulicolians infested with worms), but we do have a small number that are infested with up to 45+ worms, and this deserves comment.

Line 98

« For the anterior part of *Vetulicola* we interpret *Vermilituus gregarius* as occupying the space between the interior of the exoskeleton, and the convex surface that appears to demarcate the position of the internal anatomy »

This sentence is a bit unclear. According to most recent studies (e.g. Ou et al.) vetulicolians had some kind of spacious anterior pharyngeal chamber. That's where *Vermilituus* was probably attached. Is that what you mean ? This view is supported that the fact *Vermilituus* is found in the anterior part the animal where precisely the chamber is likely to to be found. Please rewrite this sentence accordingly.

We have rewritten as:

“

The host-symbiont relationships for *Vetulicola rectangulata* and *V. cuneata* described here support the idea of filter feeding by movement of water through the anterior section of *Vetulicola* but the attachment of robust tubes of *V. gregarius* are not consistent with the hypothesis of active pumping of water facilitated via a flexible body wall that was flexed by muscles. Our interpretation would support the presence of a space between the internal soft tissues and the inner surface of the exoskeleton to accommodate endosymbionts, and also a degree of rigidity to the exoskeleton.”

Line 111

Rather than representing post-mortem assemblages, or the result of *Vermilituus* scavenging or colonising vetulicolian carcasses, all evidence suggests that *Vermilituus* attached to the body surfaces of living vetulicolians. It is one possibility. As I wrote above it is equally plausible that *Vermilituus* was attached to carcasses (freshly dead animals).

We had already considered such an alternative in the text, but we reject it based on all the lines of evidence we discuss in detail throughout the MS. This reviewer seems to want a burden of proof for in life colonization that is just not possible in the fossil record. What we have done, aligned with other papers which discuss timing of colonization, is present all the evidence, discuss the possibilities of each and finally made our interpretation robustly and based on the evidence. We note here that Reviewer three praises us for our conservative stance and evaluation of the evidence.

They write: “Given the information at hand, I approve of the conservative stance taken regarding the type of symbiotic relationship we are seeing here, whether parasitic or commensal etc. I feel this is warranted given the mystery that surrounds both the symbiont and the host”.

Line 119

“with just 4 specimens associated with the posterior section of the most-infested specimen in our collection”

Which means that *Vermilituus* preferentially invaded the internal cavity of vetulicolians and occasionally the external surface. The reason why *Vermilituus* preferentially settled inside vetulicolians is that the inner surface was lined with a thin cuticle where it is easy to anchor. The external surface of vetulicolians was most likely thicker and harder (as you wrote in the MS). It might be relevant to

mention it here.

Yes, and we think this clearly supports our hypothesis of *in vivo* colonization.

Line 131

argue against a chance post-mortem association

Vermiltuus seems to show a random distribution which is consistent with the infestation of carcasses (see my remark above)

We have tried to be circumspect in the manuscript and offer explanations where there is some alignment, and where (mostly) there is not.

Line 133

Furthermore, the robust (possibly biomineralized) and curved tubes of *Vermiltuus gregarious* are consistent with a sessile attached ecology, but not with a motile scavenger that might have fed on vetulicolians after death.

I agree that *Vermiltuus* was a sessile organism that anchored to its host. However, I am less certain that this organism had a biomineralized exoskeleton. 3D-preservation does not necessarily mean biomineralized. For example the annulated tube of selkirkiid worms (Chengjiang Lagerstätte) is sometimes preserved in 3D although obviously not biomineralized.

We accept this and that is why we said ‘possibly biomineralized’. We do not see a need for a change here; what is most pertinent is that the tubes are rigid.

Line 146

The host-symbiont relationships for *Vetulicola rectangulata* and *V. cuneata* described here support the idea of filter feeding by active pumping of water through the anterior section of *Vetulicola*.

What suggests that vetulicolians were filter feeders is mainly the presence of a frontal opening (input) and paired lateral slits (output). The host-symbiont relationships have little to do with that and may be a supporting evidence only if you believe that *Vermiltuus* was sucked into the pharyngeal chamber (your assumed in life symbiosis hypothesis). If we consider that *Vermiltuus* lived on dead bodies, then the presence of symbionts inside vetulicolians is due to the fact that the frontal opening was big enough to let them enter inside the body. Please clarify here.

As we note above, post-mortem colonization of the host at the seabed is not a parsimonious explanation for specimens rapidly buried in an event bed.

Line 148

but would not support the hypothesis of a flexible pharynx and musculature.

Why? It sounds very unclear. The assumed pharyngeal chamber was most probably flexible and directly involved in pumping action. Please clarify.

The rigid tubes would not attach to a flexible substrate, and that is why we don’t agree with this functional interpretation of vetulicolians. Please see rewritten text under comment for line 98 above.

Line 163

Vermilituus gregarius tubes do not appear to represent the actions of animals scavenging on the Vetulicola carcass post-mortem, as encrusting tube-bearing animals are sedentary.

There is no solid evidence that Vermilituus was an encrusting organism (see above). I agree that it was not a scavenger. Most likely Vermilituus was an animal that anchored to the carcasses of vetulicolians and fed on particles released by decaying tissues (suspension feeding ?) which were probably more abundant within the internal cavity than outside Please consider this alternative option, too.

The most parsimonious explanation of the complex tubes of *Vermilituus*, inside the anterior section of vetulicolians, is that they are encrusting. The evidence for the scenario we present – in life colonization – is more rigorous and abundant across multiple lines of evidence, many of which (e.g. preferential sites on attachment) have been used frequently in the many papers, especially from the Mesozoic, analyzing such fossil associations to suggest this scenario.

Line 168

if the vetulicolian carcass was drifting through the water column for a protracted period after death

This interpretation is questionable. A nektobenthic animal as big as a vetulicolian does not stay in the water column for a long time after death, sink quite rapidly and reach the bottom within a few hours. Again, the most reasonable option is to consider Vermilituus as an animal anchoring to dead bodies.

This statement by the reviewer is not supported by any data and is an opinion. Carcasses can stay afloat for extended periods if buoyed by decay gases.

Line 202

associations of serpulids

The authors stick to their serpulid option and seem to be reluctant to explore other possibilities although reviewer 1 (first round in the revision process) had recommended them to provide more information on biofouling in extant animals. You think that “the protistan analogy does not work” (see rebuttal). However, protists are extremely frequent colonizers of extant animals and infest carcasses. Some ciliophoran protists (e.g. Acineta; see below) are relatively large and secrete a lorica (see picture). Fossilized protistans occur in Triassic myodocope ostracods as well which proves that protistans can be preserved in the fossil record. Vermilituus might have been a comparable organism. At least, please test this hypothesis.

Please think about polyps as well (see below, strobilating polyps living on an ascidian) as another hypothesis to be tested. Just open the last edition of Brusca et al. and you will see that there are many possible analogues !

As we note above, this is a misreading and we do not stick to a serpulid hypothesis. It is very clear from the supplementary file that we made a series of possible comparisons. We have chosen the analogues we bring to the narrative with careful

consideration. We have selected the most relevant taxa given the nature of the evidence and the geological time period (Cambrian).

Line 215

might have had a significant fouling effect on surfaces exposed to water flow;
See my remark above.

We think the fouling effect should be mentioned. We believe we would be criticized if we didn't point this out.

Line 219

but the additional weight of >45 individuals could have reduced the swimming efficiency of the host organism and its ability to feed

It sounds very unlikely considering the small size of parasites and the fact that they don't obstruct any openings. Please modify.

Some areas of the anterior section of several specimens have large concentrations of worms. If this area was used for feeding and respiration, as suggested in previous papers, then surely this warrants comment, given similarities to modern styles of biofouling.

Line 227

Together with the priapulid worms *Mafangscotlex sinensis* and *Cricocosmia jinningensis*, which are infested by aggregates of the small worm-like *Inquicus fellatus*.

Cong et al's excellent paper demonstrates that tiny worm-like animals could attach to the thin cuticle of priapulid worms. Since these priapulids were presumably active burrowers, the most plausible interpretation is that *Inquicus* lived on dead worms. Otherwise they would have been removed (friction with sediment). You have a very similar case here with *Vermilituus* which lived inside the carcasses of vetulicolians. I would strongly recommend to develop comparisons between these two cases. Please add a figure showing *Inquicus* and *Vermilituus* on their respective hosts.

We wish to point out that we are the authors of the Cong et al. paper. That paper documented *in vivo* infestation of worms. Those preserved in event beds were very well preserved (like our vetulicolians), and those preserved in background beds were decayed. Beyond that we don't think there is a comparison to be made.

Reviewer #2 (Remarks to the Author):

I feel the authors have made some acceptable exposition of many points raised by the reviewers. I support the publication.

We thank the reviewer for this comment

There are still some concerns worthy of careful consideration by the authors. The extraordinary Chengjiang fossils can preserve the soft tissues within the shales, which can be considered as three-dimensionally anatomical structure of organisms. as the

authors illustrated in supplementary figure 3, if the “carapace” of the animal is concrete and solid shell, there without any doubt that *Vermilituus* attached on the inner surface of *Vetulichola*. Whereas, the carapace of *Vetulichola* is preserved as membrane, or halo of disseminated soft tissues in the surrounding rock. Due to the tube of *Vermilituus* is robust and has pyritization to some extent, the small tube can be observed on multiple layers of sediment and fossil, for example, text fig 3b, 4b. The author claims that the appearance of the vermilituus tube on the inner surface of *Vetulichola*, which cannot be confirmed.

We can't agree and have shown a large number of images that show that *Vermilituus* is on the internal surfaces (including stereo-pairs that allow the reader to judge different levels in the rock slab). We have added an additional figure to show how we interpret the taphonomy, which is based on our observation of Chengjiang fossils over more than 25 years, and which are widely documented in the Chengjiang literature.

We can't accept that the anterior section (“carapace”) of *vetulicholians* is a halo of disseminated soft tissues, or simply a membrane, it is a structure with some degree of rigidity, especially if interpretations of muscle attachments to this structure are accepted (Ou et al. 2012).

Figure 4, and Supplementary Figure 2, multi- panels should be labelled with a, b, c, d.

No this is a misreading of the figure. a and b are stereo-pairs and are deliberately provided to help the reader see the surfaces of the fossils.

Reviewer #3 (Remarks to the Author):

Thanks once again for an enjoyable read. I'm glad to see this one back. I don't really have any further comments or suggestions to make on the manuscript. These interactions are so rare to capture in the fossil record, so it would be great to see this published and out there for people to read. Given the information at hand, I approve of the conservative stance taken regarding the type of symbiotic relationship we are seeing here, whether parasitic or commensal etc. I feel this is warranted given the mystery that surrounds both the symbiont and the host. Regardless, it is a very strange and interesting relationship, I don't know if I've heard of any endosymbiont that lives in a tube. But it's the Cambrian, so it would not surprise.

We thank the reviewer for these comments.

Just out of curiosity, what is the size range of *Vetulichola cuneata* and *V. rectangulata* documented from Chengjiang? And what are the sizes of the specimens that are infested with *Vermilituus*? You have listed the sizes of the tubes, but not of the host. Is it only specimens towards the larger end of the scale that are infested and do you see more tubes in the larger specimens, or is it just random? Just interested if there may be any relationship, even if the sample size is not large.

The sample size is small, but it appears to be larger specimens (longer than 7 cm) that are infested. We don't feel we can comment much more on this until a larger sample set is available.

Thanks once again and I look forward to seeing this published.

We thank the reviewer for this comment